# The Home Environment Is a Reservoir for Methicillin-Resistant Coagulase-Negative Staphylococci and Mammaliicocci

**DOI:** 10.3390/antibiotics13030279

**Published:** 2024-03-19

**Authors:** Mari Røken, Stanislav Iakhno, Anita Haug Haaland, Ane Mohn Bjelland, Yngvild Wasteson

**Affiliations:** 1Department of Paraclinical Sciences, Faculty of Veterinary Medicine, Norwegian University of Life Sciences, 1433 Ås, Norway; mari.roken@ffi.no (M.R.); anemohn.bjelland@fhi.no (A.M.B.); 2Institute of Microbiology, Norwegian Armed Forces Joint Medical Services, 2027 Kjeller, Norway; 3Previwo AS, 0454 Oslo, Norway; stanislav.iakhno@previwo.no; 4Department of Companion Animal Clinical Sciences, Faculty of Veterinary Medicine, Norwegian University of Life Sciences, 1433 Ås, Norway; anita.haug.haaland@mattilsynet.no; 5Regulations and Control Department, Animal Health, 0304 Oslo, Norway; 6Department of Bacteriology, Norwegian Institute of Public Health, 0213 Oslo, Norway

**Keywords:** methicillin resistance, antimicrobial-resistant coagulase-negative staphylococci, mammaliicocci, virulence, mobilome, home environment, pets

## Abstract

Coagulase-negative staphylococci (CoNS) and mammaliicocci are opportunistic human and animal pathogens, often resistant to multiple antimicrobials, including methicillin. Methicillin-resistant CoNS (MRCoNS) have traditionally been linked to hospitals and healthcare facilities, where they are significant contributors to nosocomial infections. However, screenings of non-hospital environments have linked MRCoNS and methicillin-resistant mammaliicocci (MRM) to other ecological niches. The aim of this study was to explore the home environment as a reservoir for MRCoNS and MRM. A total of 33 households, including households with a dog with a methicillin-resistant staphylococcal infection, households with healthy dogs or cats and households without pets, were screened for MRCoNS and MRM by sampling one human, one pet (if present) and the environment. Samples were analyzed by a selective culture-based method, and bacterial species were identified by MALDI-TOF MS and tested for antibiotic susceptibility by the agar disk diffusion method. Following whole-genome sequencing, a large diversity of SCCmec elements and sequence types was revealed, which did not indicate any clonal dissemination of specific strains. Virulome and mobilome analyses indicated a high degree of species specificity. Altogether, this study documents that the home environment is a reservoir for a variety of MRCoNS and MRM regardless of the type of household.

## 1. Introduction

Coagulase-negative staphylococci (CoNS) and *Mammaliicoccus* spp. (formerly known as the *Staphylococcus sciuri* group) are a heterogeneous group of skin and mucous membrane commensals and are also opportunistic pathogens responsible for various infections in humans and animals [1,2]. They are considered to have a lower pathogenic potential than the more virulent *Staphylococcus aureus* and *Staphylococcus pseudintermedius*. Still, CoNS cause a substantial number of infections in immunocompromised as well as in otherwise healthy patients [3]. *Staphylococcus epidermidis*, *Staphylococcus hominis* and *Staphylococcus haemolyticus* are significant contributors to septicemia in neonates, while *Staphylococcus saprophyticus* is one of the most common causative agents in urinary tract infections [4,5,6].

In addition to their opportunistic pathogenic potential, CoNS and mammaliicocci are thought to be important reservoirs for antimicrobial resistance genes (ARGs), including the *mecA* gene responsible for methicillin resistance, and mobile genetic elements associated with ARGs [1,7]. Thus, the potential for horizontal gene transfer of these genes to more pathogenic bacteria, such as *S. aureus*, is present. 

As opposed to human and animal infections caused by either methicillin-resistant *S. aureus* (MRSA) or methicillin-resistant *S. pseudintermedius* (MRSP), methicillin-resistant MRCoNS infections are not monitored in Norway. Hence, the knowledge of the occurrence of MRCoNS is largely unknown. Based on reports of high rates of MRCoNS in cases of neonatal septicemia and MRCoNS-carrying health care personnel, we can assume that Norwegian healthcare facilities make up important reservoirs for MRCoNS [6,8]. However, non-hospital environments may also serve as reservoirs for these bacteria [9]. High rates of MRCoNS have been reported in public transportation systems and humans without previous exposure to health care systems [10,11]. In a previous study on the transmission of MR *Staphylococcus* spp. (MRS) from infected dogs to their owners, we found several species of MRCoNS and MR *Mammaliicoccus* spp. (MRM) in the same home environments [12]. These findings led us to wonder if there was an association between the dogs’ infection status and the existence of MRCoNS and MRM in their homes, or if MRCoNS was a standard feature in all kinds of homes.

Thus, this study aims to gain more insight into the home environment’s role as a reservoir for MRCoNS by screening the environment, humans and pets in different households for MR bacteria. Furthermore, we examine the distribution of MRCoNS and MRM by identifying the species, sequence types and SCC*mec* elements in the home environment. Finally, we screen the bacteria for virulence genes, ARGs and mobile genetic elements.

## 2. Results

### 2.1. Isolates 

A total of 117 verified MRCoNS and MRM isolates constituted the sample material for the occurrence analysis. Of these, 103 isolates were submitted for whole-genome sequencing, of which 75 are presented in the resistome-, virulence- and SCC*mec* results. Thirty-nine *S. epidermidis*, *S. haemolyticus* and *S. hominis* isolates were included in the in silico MLST analysis, while 57 *S. epidermidis*, *S. haemolyticus*, *S. hominis* and *S. saprophyticus* isolates are presented in the mobilome analysis.

### 2.2. Occurrence of MRCoNS and MRM

All but three households (30/33) tested positive for a minimum of one species of MRCoNS or MRM (Table 1). The majority of the studied MRCoNS/MRM isolates were detected in the home environments (*n* = 107/117). Overall, MR *S. saprophyticus*, *S. haemolyticus*, *S. hominis* and *S. epidermidis* were the most prevalent species in the households. Additionally, some species appeared to be more prevalent in specific households, as follows: MR *S. epidermidis* (MRSE) in homes with infected dogs or healthy cats and MR *S. hominis* in homes with infected and/or healthy dogs. Households with infected dogs had a higher mean number of different species (3.13) compared to those with healthy pets (1.79) and without pets (1.55). A one-way ANOVA revealed that there was a statistically significant difference in the mean numbers of species between the household groups (F (2.30) = (3.487) *p* = 0.04). The Tukey’s HSD test found that this difference was between the homes with infected pets and those without pets (*p* = 0.03). 

As shown in Table 1, six humans and three dogs carried MRCoNS. Two of these dogs suffered from MRSP infections, in addition to carrying MRSE, while the third dog (household H) had an MRSE infection. Their three owners tested positive for MRSE. Two owners carried MR *S. haemolyticus*, one owner of an infected dog and one owner of a healthy cat. One owner of a healthy dog carried MR *S. warneri*. None of the humans in the household without pets tested positive for MRCoNS, while all but one of the eleven home environments tested positive for at least one species of MRCoNS.

### 2.3. Antimicrobial Resistance

Forty-one of the seventy-five isolates were multi-resistant, expressing resistance to three or more classes of antimicrobials. *S. cohnii* ssp. *cohnii*, *S. hominis* and *S. haemolyticus* were the most resistant species, expressing resistance to means of 4.3, 3.7 and 3.6 classes, respectively (Figure 1). Following resistance to anti-staphylococcal beta-lactams, resistance to macrolides (erythromycin) and fusidanes (fusidic acid) was most frequently observed (Table 2). Worth noticing is the large proportion of *S. hominis* isolates that expressed no phenotypic resistance to cefoxitin and amoxicillin clavulanic acid. The resistome analysis supported the phenotypical resistance profile, displaying a high prevalence of genes conferring resistance to erythromycin and fusidic acid among the MRCoNS and MRM (Table 3) and further confirmed that all sequenced MRCoNS and MRM possessed the *mecA* gene. Despite the presence of *mecA*, we observed a variable phenotypic expression of resistance to beta-lactams among the isolates (Table 2). This was particularly evident for *S. hominis*, with 1/11 isolates being phenotypically susceptible to oxacillin and 5/11 isolates being susceptible to cefoxitin. Furthermore, we observed phenotypic susceptibility to amoxicillin–clavulanic acid among six MRCoNS and MRM species: *S. cohnii* ssp. *cohnii* (2/3), *S. epidermidis* (8/12), *S. haemolyticus* (2/16), *S. hominis* (6/11), *S. warneri* (1/5) and *M. vitulinus* (*n* = 1). In addition to *mecA*, the beta-lactamase-encoding *blaZ* gene was present in 16 of the 20 amoxicillin–clavulanic-acid-susceptible isolates. 

### 2.4. SCCmec Cassettes and Sequence Types

Table 4 shows the predicted SCC*mec* cassettes based on detected genes and best homology in the sequenced MRCoNS and MRM isolates. For four isolates (one *S. epidermidis*, one *S. haemolyticus* and two *S. hominis*), the *SCCmec* prediction based on genes deviated from the prediction based on homology. In these cases, the SCC*mec* cassettes were reported based on the prediction of the genes. The *S. epidermidis* isolates were assigned three types (II (2a), III (3A) and IV (2b)) in addition to one non-typeable (NT) isolate. In just one household, the same SCC*mec* cassette was predicted in isolates of different species: an *S. epidermidis* and an *S. saprophyticus* isolate of type III (3A). The *SCCmec* elements of a substantial number of isolates (43/75) were non-typeable. Eleven of the fourteen NT *S. haemolyticus* isolates showed best homology with SCC*mec* type V but missed either the *ccr*C1 or *mec* class C2, or both. Three of the NT *S. saprophyticus* isolates had the best predicted homology to SCC*mec* III (3A) but missed either *ccr*A3 or both *ccr*A3 and *ccr*B3, while two additional *S. saprophyticus* isolates shared the best homology with SCC*mec* I (1B) but missed *ccr*A1 and *ccr*B1. 

The sequence types (STs) were predicted by in silico MLST for 25 of the 39 *S. epidermidis*, *S. haemolyticus* and *S. hominis* isolates. The 12 *S. epidermidis* isolates were assigned to 10 different STs (5, 35, 57, 81, 130, 218, 224, 332, 640 (*n* = 2) and 679) in addition to 1 non-typeable isolate. The two MRSE ST640s were from different households. Nine of sixteen *S. haemolyticus* were typed to seven STs (1, 3 (*n* = 2), 30, 42, 49, 52 (*n* = 2) and 56). Isolates with identical STs were from different households. The remaining *S. haemolyticus* isolates had combinations of allelic profiles not reported earlier. This was also the case for 6 of the 11 *S. hominis* isolates, while the remaining 5 were typed to two different STs (1 and 18), all of which were from different households.

### 2.5. Household Analysis of Human and Pet Isolates

In households with infected pets (C, F and H), the dogs and owners concurrently tested positive for MRSE (Table 1). In household H, the dog, owner and home environment tested positive for MRSE ST640 with identical susceptibility profiles, resistance genes and SCC*mec* elements [12]. In contrast, the dog and owner in household C carried two different STs (MRSE ST679 and ST130), presenting different resistance genes and SCC*mec* types (Appendix A). MRSE isolates with identical susceptibility profiles to the dog isolate were detected in the bathroom and kitchen. However, no isolates similar to the MRSE found on the owner were recovered from the home environment. We observed similar findings in household F, in which the owner and dog carried MRSE with different STs (ST218 and ST5) and SCC*mec* cassettes (III (3A) and IVa (2B)). Contrary to household C, we recovered only isolates with identical susceptibility profiles to the human isolate from the home environment. In household D, MR *S. haemolyticus* ST42 and ST1 were recovered from the owner, while the dog tested negative for these. Only the MR *S. haemolyticus* ST42 was detected in the home environment.

Two owners of healthy pets carried MRCoNS. The first owner tested positive for a MR *S. warneri* possessing a SCC*mec* cassette V (5C2 and 5). An MR *S. warneri* with an identical SCC*mec* cassette was recovered from the household’s kitchen. The second owner carried an MR *S. haemolyticus* with an NT SCC*mec* cassette and ST. An MR *S. haemolyticus* ST52 with different resistance genes was recovered from the home environment.

### 2.6. Virulence Genes

The hits in the virulence gene database are presented in Appendix A, and the respective *Ha* scores for each virulence gene are shown. The exfoliative toxin-encoding gene *etc* was detected in all sequenced isolates. Furthermore, we observed a high frequency of phenol soluble modulin-encoding genes, thermonuclease-encoding *nuc* genes and siderophore-encoding genes in most MRCoNS isolates. Overall, the MRCoNS and MRM showed a high degree of species specificity in the virulence gene analysis, apart from *S. haemolyticus* and *S. hominis*, which clustered together (Figure 2). We detected a high occurrence of genes involved in adherence in the *S. epidermidis* isolates (*atl*, *ebh* and *sdr* genes). In addition, we observed two subpopulations of *S. epidermidis* isolates based on the presence of Type VII secretion-associated genes (Appendix A). The tendency of two subpopulations was also evident among the *S. haemolyticus* isolates. Based on *Ha* scores of several capsular polysaccharide synthesis enzymes involved in immune evasion (*cap* genes), the group with high *Ha* scores consisted of six isolates, of which three were recovered from humans and two were of environmental origin. Isolates with low *Ha* scores for *cap* genes were solely detected in the environment.

Subgrouping based on the *Ha scores* of *cap* genes was also observed among the *S. hominis* and *S. saprophyticus* isolates. The MRM had, in general, few hits with high *Ha* scores, apart from the *etc* gene and the *capO* and *capP* genes.

### 2.7. Mobilome Analysis

The four most prevalent species found in the households, *S. saprophyticus*, *S. epidermidis*, *S. haemolyticus* and *S. hominis*, were included in the mobilome analysis. Three common gene clusters encoding an IS6 family transposase, the competence protein ComGC and an uncharacterized SPBc2 prophage-derived protein YoqJ (annotated “Common” in Figure 3) were identified in all the isolates. Otherwise, the mobilomes were mainly species-specific (Figure 3). We observed a few examples of similar gene sequences in different species at the household level. For instance, *S. haemolyticus* and *S. saprophyticus* isolates from the same household carried phage major capsid protein-encoding genes and phage portal protein-encoding genes with 76.8% and 80% amino acid identities, respectively. In addition, a site-specific tyrosine-type recombinase/integrase was shared by *S. epidermidis* and *S. saprophyticus* in two households, and an IS256-like transposase was shared by *S. haemolyticus* and *S. hominis* in two other homes. 

## 3. Discussion

MRCoNS are opportunistic pathogens prevalent in hospital environments, often due to their hardy nature, ability to form biofilms and resistance to antimicrobials [13]. The newly described genus *Mammaliicoccus* shares many properties with staphylococci, like habitat and methicillin resistance [14]. In a former study of the dissemination of clinical MRS in households with infected pets [12], we detected a broad range of MRCoNS and MRM in the home environments and from pets and their owners. To follow up on this observation, we decided to screen different categories of households for the presence of MRCoNS and MRM. To our surprise, MRCoNS and MRM were nearly ubiquitous in the home environments regardless of the presence of pets or health status. The finding of *S. epidermidis*, *S. haemolyticus*, *S. hominis* and *S. saprophyticus* as the predominant species in the households is reasonable since these species are known as skin commensals in humans. However, they can also cause infections, and the frequent occurrence of methicillin-resistant isolates in home environments is noteworthy. To our knowledge, the home environment has previously not been described as a reservoir for MRCoNS and MRM.

The skin, skin glands and mucous membranes of mammals are considered the main habitats for CoNS [15]. However, in most of the households studied, both the human and the pet tested negative for MRCoNS/MRM while the bacteria were present in the home environment. The absence of MRCoNS/MRM in humans and pets may reflect that the sampling sites in the humans and pets were not optimal for detecting some of the CoNS/mammaliicoccal species. For instance, *S. saprophyticus* is a frequent colonizer of the perineal region, rectum and urethra in humans, and *S. hominis* and *S. haemolyticus* are often isolated from axillae and pubic areas high in apocrine glands [1]. These sites were not included in the sampling procedures. On the other hand, *S. epidermidis* is a common human, canine and feline nasal mucosa colonizer [16,17]. Therefore, we find it peculiar that we identified relatively few carriers of MRSE, considering that MRSE was present in around one-third of the households. An explanation may be that we only sampled one human member in each household, thus missing possible carriers of the MRCoNS and MRM. Another factor contributing to the high number of MRCoNS/MRS in the households could be that the bacteria had been introduced via visitors, soil or other external sources. 

Carriage of MRCoNS in pets was exclusively found in infected dogs. The owners of the three MRSE-positive dogs all tested positive for MRSE. Interestingly, the isolates from the dogs and owners differed in two of the cases, indicating a diversity of MRCoNS not only between households but also within the household. Moreover, we observed that homes with infected pets had a large diversity of MRCoNS species recovered from the home environment. Five of the eight dogs in this group had been treated with beta-lactam antimicrobials within the past three months before sampling, two of which had undergone antimicrobial treatment several times during the past year. The carriership and the diversity may reflect the MRCoNS’s and MRM’s competitive advantage when exposed to beta-lactam antimicrobials. Furthermore, five dogs in this group had been hospitalized within the past twelve months, and two owners were human health care workers. Hence, it is not unlikely that the pets or owners have been exposed to MRCoNS/MRM in these environments and transmitted them further to their home environment. Still, MRCoNS and MRM were present in many households where neither humans nor dogs had been in contact with health care facilities, again emphasizing that MRCoNS and MRM are indeed found outside clinical environments. 

The phenotypic resistance analysis revealed that just over 50% of the MRM and MRCoNS were multidrug-resistant. This is consistent with previous reports on CoNS and MRCoNS in non-clinical settings [9,18]. Mobile genetic elements play a central role in spreading ARGs among bacteria [19]. Considering that MRCoNS constitute reservoirs for ARGs, we conducted a mobilome analysis primarily to investigate whether the most prevalent species in the households had mobile genetic elements in common, which could indicate genetic exchange at the household level. Nonetheless, the detected mobile genetic elements displayed mainly a species-specific profile rather than a household-related pattern. This could indicate that mobile genetic elements are not easily transmitted between different staphylococcal species. However, it must be emphasized that this analysis is based on short-read data. To gain further insight in the ARGs’ location relative to the mobile genetic elements, it would be necessary to combine short-read and long-read data. 

The inconsistent phenotypic expression of resistance to cefoxitin among the MRCoNS isolates was noteworthy. EUCAST and CLSI operate with different zone diameters when assessing cefoxitin resistance. By following the CLSI breakpoints rather than the EUCAST breakpoints, eight of the ten cefoxitin susceptible CoNS isolates would have been classified as resistant. On the other hand, two of the MRSE isolates would have been reported susceptible to cefoxitin. According to the EUCAST guidelines, cefoxitin should be used when screening for methicillin resistance in CoNS [20]. However, CLSI emphasizes that the cefoxitin disk diffusion test may not perform reliably in detecting methicillin resistance for all CoNS species (e.g., *S. haemolyticus*) [21]. Although cefoxitin is the recommended agent for most CoNS when screening for methicillin resistance, our results show that oxacillin is more reliable than cefoxitin for the purpose.

MRS are considered resistant to most beta-lactam agents, i.e., penicillins, beta-lactam combination agents and cephems, except for ceftaroline [22,23]. However, we observed a high frequency (20/75) of amoxicillin–clavulanic-acid-susceptible isolates. Sixteen of the susceptible isolates carried *blaZ*, which encodes a beta-lactamase that inactivates amoxicillin. Admittedly, this could be due to the lack of official breakpoints for amoxicillin–clavulanic acid disk diffusion. Still, six of these isolates were susceptible to cefoxitin, thus demonstrating that even if the *mecA* gene is present, it is not necessarily expressed towards cefoxitin and amoxicillin–clavulanic acid in vitro.

We could not predict STs or SCC*mec* cassettes for most MRCoNS/MRM isolates. The pubMLST database only contains data for *S. epidermidis*, *S. haemolyticus*, *S. hominis* and *S. chromogenes*, and the missing ST identifications may be due to a lack of characterized environmental genomes in the database. The high proportion of non-typeable SCC*mec* cassettes is consistent with previous reports [24,25]. In many cases, the cassettes shared homology with previously described SCC*mec* but lacked identifiable *ccr* genes needed to determine type. This was especially evident for the NT *S. haemolyticus* cassettes that had the best homology with SCC*mec* type V. The combination of NT SCC*mec* elements combined with non-identifiable STs demonstrates the large diversity among the staphylococcal and mammaliicoccal isolates in the home environments and the gaps in knowledge about the epidemiology/ecology of staphylococci from environmental reservoirs.

In general, CoNS and MRM are considered less virulent than *S. aureus*. Still, CoNS and MRM cause a substantial number of infections, presumably possessing virulence genes enabling them to do so. Virulence genes in CoNS and mammaliicocci are far less studied than the virulence genes of *S. aureus*. Consequently, we used a database mainly consisting of amino acid sequences from putative and known virulence factors in *S. aureus* to characterize virulence genes in our MRCoNS and MRM isolates [26]. Admittedly, this is not optimal and will cause uncertainty around the hits with low and medium *Ha* scores. We focused on the highest scores within each species. However, we cannot be certain that hits with lower scores are of limited importance. Overall, the MRCoNS and MRM displayed species-specific virulence gene patterns, apart from the ubiquitous *etc* gene. The virulence gene patterns revealed subgroups within the *S. epidermidis*, *S. haemolyticus*, *S. hominis* and *S. saprophyticus* isolates based on the presence of type VII secretion-associated genes for the former and *cap* genes for the three latter. The isolation source seemed to matter for multiple *cap* genes only in the *S. haemolyticus* isolates, as all the human isolates were in this subgroup.

## 4. Materials and Methods

### 4.1. Participants

Participants were recruited through social media and small animal clinics in Oslo and the surrounding areas. All participants signed individual consent forms and completed questionnaires regarding their professions, antimicrobial consumption and hospital admissions within the past 12 months. Thirty-three households participated in the study. Of these, eight were households with dogs diagnosed with an MRS infection; eight were households with clinically healthy dogs; six were households with clinically healthy cats; and eleven were households without pets. The inclusion criteria included cats with MRS infections. However, during the time we recruited participants for the study, no cats with MRS infections were diagnosed in our recruiting clinics. One human and one pet from each home participated in the study. The inclusion criteria for healthy pets were the following: clinically healthy pets without symptoms of infection when examined by a veterinarian. The humans sampled were healthy according to their own statements.

### 4.2. Sampling

The samples were collected in the period from October 2019 to October 2021. The same veterinarian was responsible for sampling all the household environments and the participating pets. Pets diagnosed with an MRS infection were sampled from the infection site, the perineum and the oral mucosa using nylon flocked swabs (Eswab™ 480C, Copan group, Brescia, Italy). These dogs participated parallelly in another study [12]. Healthy dogs and cats were sampled from the oral mucosa and perineum. Human participants collected swab samples from their nostrils and throats according to the instructions of the veterinarian present at the time of sampling. The home environments were sampled using cloths (Sodibox^®^ Swab cloth, Nevez, France) for swabbing of the most relevant areas such as the pets’ food bowls and sleeping areas, living room floors, bathrooms (sink faucet and hand towel) and kitchens (kitchen counter, dish towel, cloth and sink faucet). In the households without pets, the three latter locations were sampled. 

### 4.3. Culturing and Identification

The samples were cultured as described by Røken et al. [12]. Briefly, all samples were enriched overnight in Müller–Hinton (MH) broth supplemented with 6.5% NaCl. Then, 20 µL of MH broth was inoculated on Oxacillin Resistance Screening Agar Base (ORSAB, Oxoid, Basingstoke, UK) supplemented with 2 µg oxacillin and incubated for 24 h at 35 °C. Blue, blue-white and white colonies growing on the ORSAB agar were sub-cultured on 5% bovine blood agar overnight. The species were identified using Matrix-assisted laser desorption/ionization (MALDI-TOF MS) (VITEK^®^ MS, bioMérieux, Craponne, France). Isolates were tested for the presence of the *mecA* gene by PCR on a Bio-Rad T100 Thermal cycler (Bio-Rad, Hercules, CA, USA) [27]. 

### 4.4. Susceptibility Testing

Isolates were susceptibility tested against 12 antibiotics using the agar disk diffusion method [28]. The test panel included aminoglycoside (gentamicin 10 µg), amphenicol (chloramphenicol 30 µg), beta-lactams (amoxicillin–clavulanic acid 20/10 µg, oxacillin 1 µg, cefoxitin 30 µg, cefalexin 30 µg), fluoroquinolone (enrofloxacin 5 µg), fusidane (fusidic acid 100 µg), folate pathway antagonist (trimethoprim/sulfamethoxazole 1.25/23.75 µg), macrolide (erythromycin 15 µg), lincosamide (clindamycin 2 µg) and tetracycline (tetracycline 30 µg). When testing resistance to oxacillin, we used Müller–Hinton agar supplemented with 4% NaCl. Müller–Hinton plates were incubated at 35 °C for 18–20 h before reading the zone diameters. As there are no official breakpoints for amoxicillin–clavulanic acid and cefalexin, we used the breakpoints ≥25 mm for susceptible and ≤24 mm for resistant. Isolates displaying intermediate resistance to antimicrobial agents were registered as resistant in the phenotypic analysis. Isolates expressing resistance to three or more classes of antimicrobials were defined as multidrug-resistant [29].

### 4.5. Selection of Isolates

One MRCoNS/MRM from each species was included from each sampling location. Based on the phenotypic resistance profiles, species and households, isolates were selected for whole-genome sequencing (WGS). All WGS isolates went through an additional species identification checks using the Microbial Genomes Atlas (MiGA) webserver against the TypeMat database [30] (Table 5). If the species identities differed between the MALDI-TOF and TypeMat databases, we used the TypeMat output. Furthermore, all sequenced isolates were included in the resistome, virulence gene and SCC*mec* analyses. However, isolates that turned out to be redundant were excluded when presenting the results (isolates from the same household, identified as the same species with identical resistance genes, virulence genes, SCCmec elements and sequence types (STs)). Non-redundant sequenced *S. epidermidis*, *S. hominis*, *S. haemolyticus* and *S. saprophyticus* isolates were included in an additional mobilome analysis.

### 4.6. DNA Extraction, Library Preparation, Sequencing and Assembly

DNA was extracted using a modified version of the Master Pure™ Gram-Positive DNA Purification protocol (Lucigen Corporation, Middleton, WI, USA) [12]. The DNA quality was assessed by NanoDrop^®^ ND-1000 (Thermo Scientific, Wilmington, CA, USA), and the DNA quantity was determined using a Qubit fluorometer with the dsDNA Broad Range Assay kit (Invitrogen, Eugene, OR, USA). Quality-controlled DNA was submitted to the Norwegian Sequencing Centre for library preparation and sequencing. The library prep was performed in two batches using Swift Turbo 2s flex DNA library prep on batch one and Nextera DNA Flex Prep on batch two. The samples were sequenced on the Illumina MiSeq platform v3, resulting in 300 bp paired-end reads. The fastq files were quality checked using FastQC version 0.11.9. Adapters and low-quality sequences were removed using Trim Galore version 0.6.7 [31]. We used SPAdes version 3.15.3 for genome assembly [32].

### 4.7. Bioinformatics

We used ABRicate version 1.0.1 for the resistome analysis [33]. The assembled sequences were run against the CARD database with cutoff values of 80% nucleotide identity and 80% coverage [34]. We used SCC*mec*Finder v. 1.2 with default settings (nucleotide identity 90% and minimum sequence length of 60%) to type SCC*mec* elements [35]. *S. epidermidis*, *S. hominis* and *S. haemolyticus* assemblies were run against a default PubMLST scheme using MLST version 2.19.0 [36]. 

For the virulence gene analysis, we aligned a custom database containing amino acid sequences of staphylococcal virulence factors [26] against assembled staphylococcal and mammaliicoccal genomes using tblastn (v. 2.5.0) with the default settings, except for high-scoring segment pair (HSP) set to 1 and the culling limit of 1. The resulting sample/VF matrix values were expressed as *Ha* scores ranging from 0 to 1 [26]. Briefly, the scores were calculated using the following formula: *Ha* = (pident × length)/(qlen × 100)
where “pident” represents the proportion of amino acid sequence identities between the VF query and translated proteins from the bacterial genomes in this study, “length” represents the alignment length of a hit and “qlen” is the length of the query sequence drawn for each VF.

The mobilome analysis was conducted using Anvi’o bioinformatics suite version 7.1. [37]. Before the pangenome analysis, we excluded one *S. saprophyticus* from the dataset due to a high number (>2000) of partial genes. We created the Anvi’o contigs database with the “anvi-gen-contigs-database” program using Prodigal to identify open reading frames [38]. The resulting genes were associated with the functions from the NCBI’s Clusters of Orthologous Groups (COGs) database [39]. The pangenome was computed by the core Anvi’o program “anvi-pan-genome” (default settings), which in turn utilized DIAMOND [40] and MCL [41]. Metadata were integrated into the pangenome results with the “anvi-import-misc-data” program. After the pangenome visualization, we extracted all gene clusters annotated with the “Mobilome” COGs category with the “anvi-split” program for further manual inspection. We used COG annotations or an ad hoc protein web-blast search to characterize the gene clusters of the staphylococcal mobilome.

### 4.8. Statistical Analysis

We used one-way ANOVA to compare the number of different MRCoNS and MRM species between the households with infected pets, with healthy pets and without pets. Tukey’s honest significant difference (HSD) test was applied to test the pairwise difference between the three household groups. The significance level was set to 0.05.

The virulence factor (VF) matrix was transferred to R version 4.1.0 for further principal component analysis (PCA) using the “prcomp” function of the “stats” package (v. 4.0.1), followed by a visualization using the “fviz_pca” function of the “factoextra” package (v. 1.0.7).

## 5. Conclusions

In conclusion, we have documented that the home environment is a reservoir for MRCoNS and MRM regardless of the type of household and the carrier status of humans and pets. However, homes with infected pets had a larger diversity in MRCoNS and MRM species than households without pets, which might be due to the recent use of antimicrobials and contact with human and veterinary hospitals. The large diversity in SCC*mec* elements and sequence types among and within the households indicates no clonal spread of specific strains. The limited common virulomes and mobilomes indicate a high degree of species specificity rather than exchanges of genetic elements between species in the home environment.

## Figures and Tables

**Figure 1 antibiotics-13-00279-f001:**
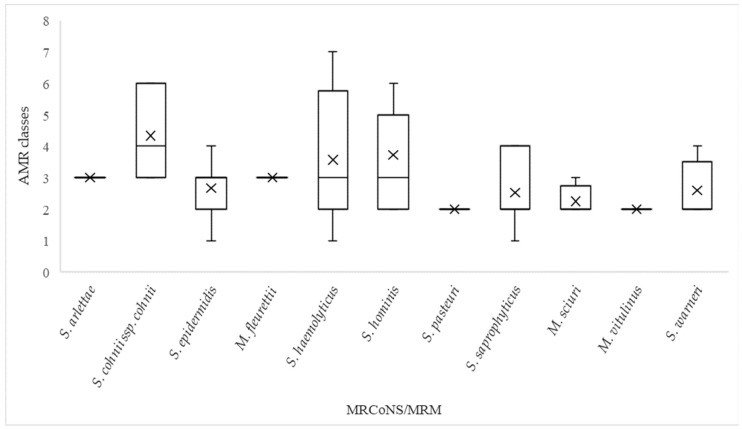
Number of phenotypic resistance classes in the MRCoNS and MRM isolates, *n* = 75. The crosses represent the mean number of antimicrobial resistance (AMR) classes, the horizontal lines represent the median number, the boxes represent the quartiles, while the whiskers represent minimum and maximum number or resistance classes.

**Figure 2 antibiotics-13-00279-f002:**
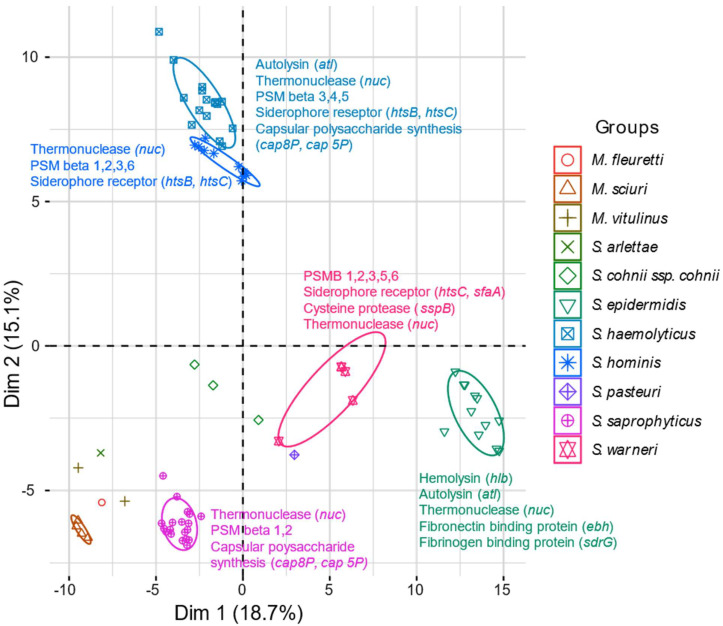
Principal component analysis plot of virulence gene similarity in MRCoNS and MRM isolates. The exfoliative toxin gene *etc* was present in all isolates. The annotations refer to the genes with the highest Ha scores among the isolates.

**Figure 3 antibiotics-13-00279-f003:**
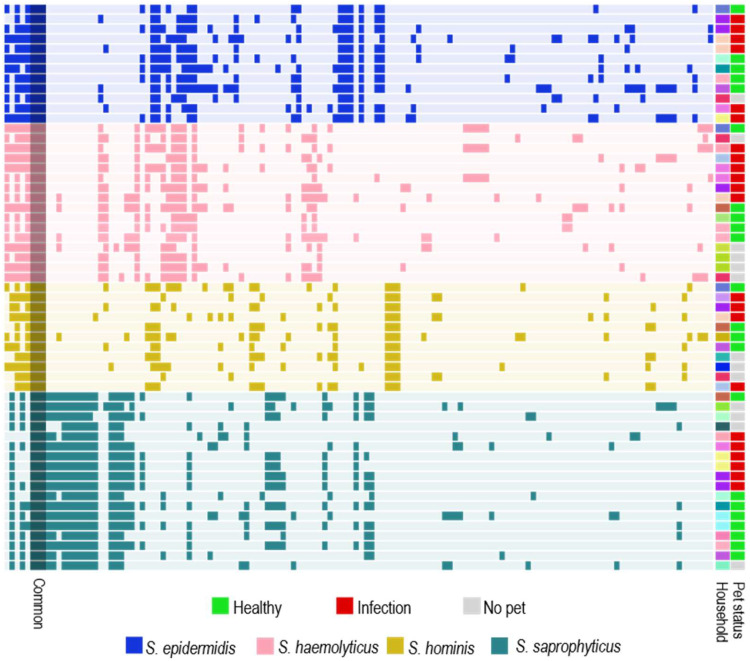
Mobilome of *S. epidermidis*, *S. haemolyticus*, *S. hominis* and *S. saprophyticus*, sorted by species. The colors in the “Household” column represent the different households the isolates were isolated from. Gene clusters identified in all isolates are annotated as “Common” and include genes encoding an IS6 family transposase, the competence protein ComGC and an uncharacterized SPBc2 prophage-derived protein YoqJ.

**Table 1 antibiotics-13-00279-t001:** Overview of antimicrobial use (AM), hospital admission and location of methicillin-resistant coagulase-negative *Staphylococcus* spp. (MRCoNS) and methicillin-resistant *Mammaliicoccus* spp. (MRM) in the 33 households. B = Bathroom; FB = Food bowl; FL = Floor; K = Kitchen; S = Sleeping place.

Status	Household	Pet AM Treatment (within Months)	Human AM Treatment (within 12 Months)	Hospital Admission (within 12 Months)	Work in Health Care	Home Environment											Human Carriage	Pet Carriage
						*S. arlettae*	*S. cohnii* ssp. *cohnii*	*S. epidermidis*	*M. fleurettii*	*S. haemolyticus*	*S. hominis*	*S. pasteuri*	*S. saprophyticus*	*M. sciuri*	*M. vitulinus*	*S. warneri*		
Household with dogs with infection	A				Yes					B, FB, K	FB, K		B, FL, K					
	B			Pet							B, FL K							
	C	Cefalexin (0–3) Polymyxin B (3–6)	Unknown agent					B, K		FB, FL	B					FB, S	*S. epidermidis*	*S. epidermidis*
	D	Cefalexin (0–3)		Pet				S		B, FL, K, S			B	FL, S			*S. haemolyticus*	
	E	Amoxicillin (0–3) Trim/sulfa (0–3)		Pet			B			K			FB	FL				
	F	Amoxicillin 0–3) Cefalexin (3–6) Enrofloxacin (3–6)		Pet. Human	Yes			K, S	B	B	FB		FL, K			FL, S	*S. epidermidis*	*S. epidermidis*
	G			Pet			FL											
	H	Amoxicillin (0–3) Fusidic acid (6–12)						K					FB, K				*S. epidermidis*	*S. epidermidis*
Household with healthy dogs	1									FL, S	FB		FL, K					
	2																	
	3		Clindamycin	Human							S					K	*S. warneri*	
	4	Amoxicillin–clav (0–3)	Penicillin			S												
	5	Fusidic acid (0–3)	Chloramphenicol	Pet	Yes								B, FL, S					
	6																	
	7			Human							B		FB, FL					
	8		Erythromycin	Human				B, FB, FL, S, K			B	FL	FB					
Household with heathy cats	9		Pivemecillinam					B, FL		K	B, FB, S							
	10	Amoxicillin (0–3)						FB					FL					
	11							FB					FB, S					
	12												FL					
	13									FL, S								
	14							B		FL			FL				*S. haemolyticus*	
Household without pets	I				Yes								B					
	II				Yes										FL, K			
	III												B	K				
	IV		Penicillin				FL						B, K					
	V												FL					
	VI				Yes					B								
	VII		Penicillin															
	VIII			Human						B, FL					FL			
	IX		Unknown agent	Human							K							
	X										FL, K							
	XI		Tetracycline					K		FL, K	FL			K		B		

**Table 2 antibiotics-13-00279-t002:** Percentage of phenotypically expressed resistance in the 75 sequenced MRCoNS and MRM isolates. A darker shade represents a higher percentage. Gen: Gentamicin; Chl: Chloramphenicol; Oxa: Oxacillin; Cfox: Cefoxitin; AmCl: Amoxicillin–clavulanic acid; Clex: Cefalexin; Enr: Enrofloxacin; T/S: Trimethoprim sulfamethoxazole; Fus: Fusidic acid; Cli: Clindamycin; Ery: Erythromycin; Tet: Tetracycline.

		Antimicrobial Agent											
Species	*n*	Gen	Chl	Oxa	Cfox	AmCl	Clex	Enr	T/S	Fus	Cli	Ery	Tet
All isolates	75	20	5	99	88	73	95	12	21	51	21	51	20
*S. arlettae*	1			100	100	100	100					100	100
*S. cohnii* ssp. *cohnii*	3		33	100	100	33	100		33	33	67	100	67
*S. epidermidis*	12	8		100	100	33	83	8	17	50	17	50	17
*M. fleuretti*	1			100	100	100	100			100	100		
*S. haemolyticus*	16	50	6	100	88	88	100	38	25	25	31	50	31
*S. hominis*	11	46		91	55	46	82	18	46	55	27	64	27
*S. pasteuri*	1			100	100	100	100					100	
*S. saprophyticus*	19		11	100	95	100	100		16	53	5	58	11
*M. sciuri*	4			100	100	100	100			100	25		
*M. vitulinus*	2			100	50	50	100			100			
*S. warneri*	5	20		100	100	80	80		20	80	20	20	

**Table 3 antibiotics-13-00279-t003:** Percentage of the 75 isolates testing positive for antimicrobial resistance genes (ARGs). All numbers are percentages of the number of isolates shown in column 2. A darker shade represents a higher percentage.

Antimicrobial Class		ID	*S. arlettae*	*S. cohnii* ssp. *cohnii*	*S. epidermidis*	*M. fleurettii*	*S. haemolyticus*	*S. hominis*	*S. pasteuri*	*S. saprophyticus*	*M. sciuri*	*M. vitulinus*	*S. warneri*
	ARG	All Isolates *n* = 75	1	3	12	1	16	11	1	19	4	2	5
Aminoglycoside	*aac(6′)-le/aph(2″)-la*	20			8		56	46					
	*ant(4′)-lb*	13		67	25		31						
	*aph(3′)-IIIa*	4					19						
	*sat4*	5			8		19						
Amphenicol	*cat*	4		33						11			
	*catA*	1					6						
Beta-lactam	*arl*	1	100										
	*blaZ*	63		33	100		94	82		32			80
	*mecA*	100	100	100	100	100	100	100	100	100	100	100	100
Folate pathway antagonist	*dfrC*	27			92			9		21			80
	*dfrG*	8			8		31						
Fosfomycin	*fosB6*	4						27					
	*fosD*	1									25		
Fusidane	*fusB*	29			50		25	36		21			80
	*fusC*	7			8		6	27					
	*fusD*	25								100			
	*fusF*	4		100									
Macrolide, lincosamide, streptogramin	*ermC*	13			8		31	18		5			20
	*lnuA*	15		67				27		32			
	*mphC*	29		100	17		38	9	100	47			
	*msrA*	43	100	100	42		38	42	100	46			20
	*salA*	5									100		
	*vgaA*	4						8		9			
	*vgaALC*	5					19						
Multidrug	*mgrA*	61		33	17		100	92	100				100
	*norA*	24			100			8	100				80
Pseudomonic acid	*mupA*	5					13	17					
Quaternary ammonium compounds	*qacA*	41		33	42		69	91		11			40
	*qacB*	3		33					100				
Tetracycline	*tetK*	20	100	67	17		38	18		11			
	*tetL*	3					13						

**Table 4 antibiotics-13-00279-t004:** Predicted SCC*mec* elements based on detected genes in the sequenced MRCoNS and MRM isolates.

Species ID	II (2A)	III (3A)	IV (2B)	IVa (2B)	IVd (2B)	IVc (2B)	V (5C2 and 5)	VIII (4A)	Non-Typeable
*S. arlettae*			1						
*S. cohnii* ssp. *cohnii*									3
*S. epidermidis*	1	1	2	6	1				1
*M. fleurettii*									1
*S. haemolyticus*							1	1	14
*S. hominis*								4	7
*S. pasteuri*									1
*S. saprophyticus*		10							9
*M. sciuri*									4
*M. vitulinus*									2
*S. warneri*						3	1		1

**Table 5 antibiotics-13-00279-t005:** Criteria for including isolates in the different analyses.

Analysis	Isolates Included in the Analysis	Isolates Presented in the Results Section
*mecA* PCR	All cultured isolates	
Species identification (MALDI-TOF MS)	All cultured isolates	
Susceptibility testing	All cultured isolates	Non-redundant WGS isolates
Whole-genome sequencing (WGS)	Non-redundant isolates based on phenotypical resistance profiles, species and household	
Additional species identification (in silico, TypeMat)	All WGS isolates	
Resistome analysis	All WGS isolates	Non-redundant WGS isolates
Virulence gene analysis	All WGS isolates	Non-redundant WGS isolates
SCC*mec* typing	All WGS isolates	Non-redundant WGS isolates
Sequence typing (in silico MLST)	All WGS *S. epidermidis*, *S. hominis* and *S. haemolyticus* isolates	Non-redundant WGS *S. epidermidis*, *S. hominis* and *S. haemolyticus* isolates
Mobilome analysis	All whole genome-sequenced *S. epidermidis*, *S. haemolyticus*, *S. hominis* and *S. saprophyticus* isolates	Non-redundant *S. epidermidis*, *S. haemolyticus*, *S. hominis* and *S. saprophyticus* isolates

## Data Availability

The sequences included in the analyses are available at https://www.ncbi.nlm.nih.gov/bioproject/?term=PRJNA856113 (accessed on 6 July 2022).

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
