# Peer review of "The Home Environment Is a Reservoir for Methicillin-Resistant Coagulase-Negative Staphylococci and Mammaliicocci"

_antibiotics, 2024, doi:10.3390/antibiotics13030279_

Round 1

Reviewer 1 Report

Comments and Suggestions for Authors

This is a nicely written study that presented important public health findings. However, I have the following major queries:

1.      I was wondering why data on the plasmids and transposons of the AMR genes were not presented. Because some AMR genes are plasmid-bound or transposon-linked. These will give better information about the potential transmission of some AMR genes across hosts in the same households.

2.      I think species-specific phylogenomic analyses of the strains are relevant. Especially, the SNP difference of strains with similar AMR genes and ST or clustering could confirm the transmission pattern of the species across different hosts.

3.      Were there no S. aureus strains from this study? Perhaps, if there are MRSA strains, the SCCmec types could be analyzed with respect to those present in the CoNS. This is because it has been opined that SCCmec of the mecA gene in MRSA originated from MRCoNS through horizontal gene transfer. Especially if the strains (MRSA and MRCoNS) are in the same niche.

Some minor comments:

1.      Could you expand the abstract to accommodate the IMRaD: Introduction, Methods, Results, and Discussion/Conclusion?

2.      The introduction of the article is too brief (short). Could you add one more paragraph after line 37? In this section, indicates the relevance of CONS and mammaliiococci as reservoirs and vectors of AMR genes (including critical AMR genes) to S. aureus.

3.      In line, expand the MR as the first instance of us… to methicillin-resistant (MR)

4.      In Line 46, kindly change “In a previous study on transmission of MR Staphylococcus spp. (MRS)

5.      In line 70, change “tended to be more prevalent in specific households:” to “appeared tended to be more prevalent in specific households as follows:

Comments on the Quality of English Language

Good. But try to cross-check the entire text to avoid typo, syntax or grammatical errors. 

Reviewer 2 Report

Comments and Suggestions for Authors

“The home environment is a reservoir for methicillin-resistant coagulase-negative staphylococci and mammaliicocci” by Røken et al. 

The paper is on a study to explore the home environment as a reservoir for methicillin-resistant coagulase-negative staphylococci and methicillin-resistant mammaliicocci. They conclude that the home environment is a reservoir for both these species regardless of the type of household and the carrier status of humans and pets. The paper represents a lot of work and detailed study of the subject. The result is very significant for knowledge of infection and antibiotic resistance related to these bacteria. One interesting observation is that in many households both the humans and the pets tested negative for the bacteria even though the bacteria were present in the home environment. The paper is very well-written. I have a few additional minor comments on the manuscript.

Line 30: “but are also” It is my understanding that “but also” should be preceded somewhere in the sentence by “not only”, or, you can change it to “and are also”

Line 33: Change “in immunocompromised but also in” to either “not only in immunocompromised but also in” or “in immunocompromised as well as in”

Line 34: “For instance” Not clear what the fact in this sentence is an example of. The immediately preceding sentence, or the sentence before that?

Line 38: “MR S. aureus” Need to write the full form the first time it is mentioned such as, “methicillin resistant (MR) S. aureus”.

Line 70-71: “MR S. epidermidis (MRSE) in homes with infected dogs and healthy cats and MR S. hominis in homes with infected and healthy dogs.” This sentence is unclear and probably incorrect. It suggests that the same home had both healthy and infected pets. Please make the sentence clear.  Possibly can replace “and” with “or” or “and/or”

Line 79: “one with MRSE in addition to carrying MRSE.” Not clear what this means.

Line 86 Table 1 and other places: Not clear why there is no data on households with infected cats. I could not find it mentioned anywhere but is important to comment.

Line 111: Figure 1 legend does not have sufficient information. What do the boxes represent? What do the directional error bars represent?

Line 114-117: The significance of the colors in Table 2 has not been described.

Line 118 Table 2: Not clear how 6% is calculated. 5/75 =7%

Line 120 Table 3: It will help if it is mentioned in the title or as a footnote that all numbers are percentages of the number of isolates shown in row 2. If done that way, writing % in row 3 column 3 is not necessary. 

Line 205-208: The 2nd last column has many more colors than those mentioned in the figure legend. Please mention what those colors represent.

Line 276: “MRS are considered resistant to other beta-lactam agents, i.e., penicillins, beta-lactam combination agents, cephems (except for ceftaroline) and carbapenems.” Can you cite a reference for this statement?

Comments on the Quality of English Language

English is fine.

Round 2

Reviewer 1 Report

Comments and Suggestions for Authors

They have satisfactorily addressed all the queries.